# Anti-*Acanthamoeba* Activity of Brominated Sesquiterpenes from *Laurencia johnstonii*

**DOI:** 10.3390/md16110443

**Published:** 2018-11-11

**Authors:** Sara García-Davis, Ines Sifaoui, María Reyes-Batlle, Ezequiel Viveros-Valdez, José E. Piñero, Jacob Lorenzo-Morales, José J. Fernández, Ana R. Díaz-Marrero

**Affiliations:** 1Instituto Universitario de Bio-Orgánica Antonio González (IUBO AG), Centro de Investigaciones Biomédicas de Canarias (CIBICAN), Universidad de La Laguna (ULL), Avda. Astrofísico F. Sánchez, 2, 38206 La Laguna, Tenerife, Spain; sara.garciadv@uanl.edu.mx; 2Facultad de Ciencias Biológicas, Universidad Autónoma de Nuevo León (UANL), Av. Pedro de Alba s/n, 66450 San Nicolás de los Garza, Nuevo León, Mexico; jose.viverosvld@uanl.edu.mx; 3Instituto Universitario de Enfermedades Tropicales y Salud Pública de Islas Canarias, Universidad de La Laguna, Av. Astrofísico Francisco Sánchez s/n, 38206 La Laguna, Tenerife, Spain; ines.sifaoui@hotmail.com (I.S.); mreyesba@ull.edu.es (M.R.-B.); jpinero@ull.edu.es (J.E.P.); jmlorenz@ull.edu.es (J.L.-M.); 4Departamento de Química Orgánica, Universidad de La Laguna (ULL), Avda. Astrofísico F. Sánchez, 2, 38206 La Laguna, Tenerife, Spain

**Keywords:** brominated sesquiterpene, marine natural products, *Laurencia johnstonii*, johnstane, 3-bromojohnstane, anti-amoeboid activity, *Acanthamoeba*

## Abstract

Focused on our interest to develop novel antiparasistic agents, the present study was aimed to evaluate the biological activity of an extract of *Laurencia johnstonii* collected in Baja California Sur, Mexico, against an *Acantamoeba castellanii* Neff strain. Bioassay-guided fractionation allowed us to identify the amoebicidal diastereoisomers α-bromocuparane (**4**) and α-isobromocuparane (**5**). Furthermore, bromination of the inactive laurinterol (**1**) and isolaurinterol (**2**) yielded four halogenated derivatives, (**6**)–(**9**), which improved the activity of the natural sesquiterpenes. Among them, the most active compound was 3α-bromojohnstane (**7**), a sesquiterpene derivative which possesses a novel carbon skeleton johnstane.

## 1. Introduction

Free-living amoeba (FLA) are widely distributed protozoa in the environment [1,2,3]. These parasites present a life cycle with two different stages: the trophozoite and the resistant phase, the cyst. Among FLA, *Acanthamoeba* genus [4] has been isolated from air, soil, water, contact lenses, air conditioning units, clinical samples, among others [5]. These parasites are able to cause pathologies in humans such as Granulomatous Amoebic Encephalitis (GAE) and Amoebic Keratitis (AK) [1,2,3,5]. Regarding *Acanthamoeba* infections, an early diagnosis is crucial to achieve a successful treatment [3,6]. Antimicrobial chemotherapy is the most widely used method for treating *Acanthamoeba*-caused infections. Pentamidine, azoles, sulfonamides, and possibly flucytosine, are among the most frequently used drugs in successfully treated cases of GAE, whereas topical chlorhexidine or polyhexamethylene biguanide appear to be the most effective option in cases of AK [7]. However, the existing therapies are not fully effective against these organisms mainly due to the existence of the cyst phase, and also due to the presence of strains that are resistant to the currently used anti-amoebic drugs [2,3,8].

In the past, natural products have been used for the treatment of different parasitic diseases. Artemisinin, quinine, amphotericin B, and ivermectin are examples of important antiparasitic compounds isolated from plants and microorganisms [9,10]. From 1981 to 2014, 15 small molecules were approved as antiparasitic drugs; among them, two were obtained from natural sources and five are the result of semisynthetic derivatives of natural products [11], however, none of them came from marine sources. Many other natural products of diverse molecular structures have revealed antiparasitic potency in the laboratory and, thus, represent interesting lead structures for the development of new and urgently needed antiparasitic agents.

The genus *Laurencia* is one of the richest sources of novel secondary metabolites among red algae [12,13,14,15]. Although the chemistry of *Laurencia* species has been exhaustively investigated, the biological activity of the isolated secondary metabolites has not been studied in a systematic way. Furthermore, *Laurencia* metabolites have been revealed to possess antiparasitic properties against a number of parasites and their vectors. However, no activity has been previously reported against *Acanthamoeba* [10,16].

Focused on our interest to develop novel lead structures for the development of antiparasistic agents, the present study was aimed to evaluate the biological activity of an extract of *Laurencia johnstonii* against an *Acanthamoeba castellanii* Neff strain. To the best of our knowledge, johnstonol [17] and prepacifenol epoxide [18] are the only sesquiterpenes isolated from specimens of *Laurencia johnstonii*, both with a chamigrene carbon skeleton. In this paper, we report the isolation of the known brominated metabolites laurinterol (**1**), isolaurinterol (**2**)**,** aplysin (**3**), and the antiparasitic diastereoisomers α-bromocuparane (**4**) and α-isobromocuparane (**5**), with cyclolaurane, laurane and cuparane backbones. (Figure 1). In addition, transformation of (**1**) and (**2**) has yielded four structural derivatives (**6**–**9**) which improved the antiparasitic activity with respect to the natural products.

## 2. Results and Discussion

### 2.1. Isolation and Identification of Natural Brominated Sesquiterpenes

*Laurencia johnstonii* was collected off the coast of the Gulf of California, Mexico. Clean and dry specimens were extracted in ethanol to give a crude extract which showed moderate anti-*Acanthamoeba* activity with an IC_50_ value of 125.14 ± 4.5 µg/mL. The bioassay-guided fractionation of the ethanolic extract by Sephadex LH-20 led us to an active fraction (SF3). This fraction was further chromatographed over a silica gel column using a step-gradient from *n*-hexane to EtOAc to give seven subfractions. The most active was SF3.1 (*n*-hexane fraction) with an IC_50_ value of 101.29 ± 0.2 µg/mL. After a separation on a silica gel open column eluted with mixtures of *n*-hexane:EtOAc, fraction SF3.1 yielded aplysin (**3**) [19], and the active stereoisomers α-bromocuparane (**4**) and α-isobromocuparane (**5**) [20], which showed activity against *Acanthamoeba castellanii* Neff with IC_50_ values of 90.68 ± 1.5 and 64.25 ± 1.5 µg/mL, respectively. Additionally, laurinterol (**1**) [21], and isolaurinterol (**2**) [22] were isolated from the inactive fraction SF3.2. The NMR, mass spectrometry, and optical rotation data of (**1**)–(**5**) were compared with those previously reported in the literature to confirm their structures (Appendix A).

### 2.2. Derivatization of Laurinterol *(**1**)* and Isolaurinterol *(**2**)*

Halogenation is a common biosynthetic strategy found in marine organisms as is the case of *Laurencia* species [23]. Halogen-containing natural products display a wide range of biological activities; therefore, they are interesting structures for medicinal chemistry studies. The presence of halogen substituents in many natural products also enhances their biological activities [24]. In order to evaluate structure−activity relationships of laurinterol and its congeners in antiparasitic assays, bromination of (**1**) and (**2**) were carried out to obtain their brominated derivatives. Hence, laurinterol (**1**) was dissolved in ethyl ether and treated with bromine. The reaction mixture was chromatographed on a silica gel column to obtain the brominated sesquiterpene derivatives (**6**) and (**7**) (Figure 2).

Compound (**6**) was obtained as a colorless oil. Its molecular formula was established by HR-EI-MS (High-Resolution Electron Impact Mass Spectrometry) analysis as C_15_H_18_Br_2_O by the presence of three molecular ions [M]^+^ at *m*/*z* 371.9711/373.9702/375.9677 (calcd. 371.9724/373.9704/375.9683; ratio 50:100:48), indicative of two bromine atoms and six degrees of unsaturation. The structure of (**6**) was determined by comparison of its NMR data with those for aplysin (**3**) (Table 1). The main differences observed in their ^1^H NMR spectra were the absence of one of the two aromatic singlets of (**3**), H-8 (δ_H_ 6.59), and a deshielded aromatic methyl group H_3_-15 (δ_H_ 2.49) for (**6**) compared with that of **3** H_3_-15 (δ_H_ 2.30). These values agree with the presence of a bromine atom at C-8 in the aromatic ring as well as with the information provided by mass spectrometry.

Aplysin (**3**) was easily obtained after treatment of laurinterol (**1**) with HBr (Figure 2). The specific rotations for (**1**) ([α]${}_{D}^{25}$ +17) and **3** ([α]${}_{D}^{25}$ −49) are consistent with those reported in the literature for both compounds (**1**): [α]${}_{D}^{25}$ +13 and **3**: [α]${}_{D}^{25}$ −86, for which the absolute configuration was stablished by X-ray crystallography [21,25]. According to this, and supported by consistent chemical shifts and coupling constants of H_3_-12, H_3_-13, H_3_-14 and H-3 between **6** and aplysin (**3**) (Table 1), the absolute configurations of **6** are 1*S*, 2*S*, and 3*S*. Caccamese and Rinehart identified **6** as bromoaplysine in 1978 by GC-MS [26]. Later, in 2010, a patented methodology for the extraction of antiobesity sesquiterpenes from *Laurencia* species [27] refers **6** as 8-bromoaplysin, however no complete NMR data has been reported for this brominated sesquiterpene.

Compound **7** was obtained as a colorless oil. The HREIMS analysis provided four molecular ions [M]^+^ at *m*/*z* 449.8840/451.8800/453.8798/455.8758 (calcd. 449.8829/451.8809/453.8789/455.8768; ratio 26:78:67:26) suggesting a molecular formula C_15_H_17_OBr_3_ and six degrees of unsaturation. The structure of **7** was established based on the analysis of its spectroscopic data and comparison with those of **6**. 1D and 2D NMR spectra evidenced three methyl, three methylenes, and two methine groups (one on sp^2^ carbon and one bearing heteroatom), in addition to five sp^2^ and two sp^3^ quaternary carbons. Similar to **6**, the pentasubstituted phenyl ring was confirmed by the presence of a singlet at δ_H_ 7.14 (H-12) and the methyl group at δ_H_ 2.53 (H_3_-15) in the ^1^H NMR spectrum (Table 2), and the HMBC correlations from H_3_-13 (δ_H_ 2.54, s) and H_3_-14 (δ_H_ 1.04, s) to a quaternary carbon at δ_C_ 92.3 (C-2), and from H_3_-14 to C-7 (δ_C_ 136.3) confirming the presence of the same dihydrofuran ring found in **6**. The main difference between both compounds was a COSY spin system from the diasterotopic protons H_2_-3 (δ_H_ 2.42/1.99), sequentially coupled with the bromomethine H-4 (δ_H_ 3.84), methylene H_2_-5 (δ_H_ 2.18/1.76), and methylene H_2_-6 (δ_H_ 2.18/1.54). HMBC correlations from H_3_-14 (δ_H_ 1.04) to C-6 (δ_C_ 32.8) and H_3_-13 (δ_H_ 2.54) to C-3 (δ_C_ 47.6) led to the connection of this substructure within the molecule and thus established the planar structure of the rearranged sesquiterpene **7** as shown in Figure 3.

NOE correlations observed from H_3_-13 to H_3_-14 and H-4, and from H-4 to the diastereotopic proton H-3β located all these protons on the same face of the molecule. Since the absolute configuration of laurinterol (**1**) has been confirmed and configuration at C-1 is conserved in **7**, it is possible to establish the absolute configuration of its chiral centers as 1*S*, 2*S*, and 4*S*.

In 1972, Yamada et al. reported the synthesis of 2,3-dimethyl-3-(5-bromo-2-methoxy-4-methylphenyl)-cyclohexane from an anisole derivative as a key intermediate to obtain aplysin (**3**) and other related sesquiterpenes [28]. Also, synthetic efforts to obtain the sesquiterpene cuparene through radical cyclization exclusively yielded 1-(1,2-dimethylcyclohexyl)-4-methylbenzene [29].

It is noteworthy that, even though chemical investigations on the prolific genus *Laurencia* have yielded over 500 sesquiterpene metabolites belonging to more than 50 carbon frameworks [16], so far none has been reported to possess that of **7**. Thus, for this carbon backbone we propose the name johnstane and the name 3α-bromojohnstane for **7** (Figure 4).

In the same way, isolaurinterol (**2**) was dissolved in ethyl ether and treated with bromine to afford **6**, and the known sesquiterpenes 8,10-dibromoisoaplysin (**8**) [30] as well as the 13-hydroxy derivative **9**, also obtained after bromination of debromoaplysinol [31].

### 2.3. Antiparasitic Effect Against Acanthamoeba Castellanii Neff Strain

As well as in the case of natural sesquiterpenes **1**–**5**, the antiamoeboid activity of the brominated sesquiterpene derivatives **6**–**9** was tested against *Acanthamoeba castellanii* Neff (Inhibitory Concentrations (IC_50_) shown in Table 3). Moreover, their toxicity (Cytotoxic Concentrations (CC_50_) at 24 h, Table 3) against murine macrophages J774.A1 (ATCC # TIB-67) at 24 h was evaluated as previously described [32,33].

As shown in Table 3, all brominated derivatives (**6**–**9**) improved the activity of the natural sesquiterpenes (**1**–**5**). Monobrominated compounds in the aromatic ring, **1**–**3**, were inactive, thus suggesting that dibromination of the aromatic moiety is relevant to obtain antiamoeboid activity. On the other hand, natural diastereoisomers **4** and **5**, which only differ in the configuration of the bromine atom attached to C-3 of cyclopentane ring, showed a differentiated activity. It is interesting to note that an α-oriented bromine substitution at C-3 in **5** increases the activity against *A. castellanii* while decreases the toxicity, a structural feature also found in compound **7**. 3α-bromojohnstane (**7**) was the most active molecule against *A. castellanii* with an IC_50_ value of 41.51 µM, and one of those with lower toxicity values against murine macrophages among the assayed compounds.

## 3. Materials and Methods

### 3.1. General Experimental Procedures

Optical rotations were measured in CH_2_Cl_2_ on a PerkinElmer 241 polarimeter (Waltham, MA, USA), by using a sodium lamp operating at 589 nm. NMR spectra were recorded on a Bruker AVANCE 500 MHz or 600 MHz, as required. NMR spectra were obtained dissolving samples in CDCl_3_ (99.9%) and chemical shifts are reported relative to solvent (δ_H_ 7.26 and δ_C_ 77.0 ppm) and TMS as internal pattern. Bruker AVANCE 600 MHz instrument is equipped with a 5 mm TCI inverse detection cryoprobe. Standard Bruker NMR pulse sequences were utilized. HR-ESI-MS (High-Resolution ElectroSpray Ionization Mass Spectrometry) and HR-EI-MS (High-Resolution Electron Impact Mass Spectrometry) data were obtained on a Waters LCT Premier XE Micromass (Manchester, UK) and VG-AutoSpec Micromass (Manchester, UK) spectrometers, respectively. IR spectra were recorded on a Bruker IFS66/S (Ettlingen, Germany) equipped with an ATR accessory using CH_2_Cl_2_ solutions. EnSpire^®^ Multimode Reader (Perkin Elmer, Walt, MA, USA) using absorbance values of Alamar Blue^®^ reagent. TLC (Thin layer chromatography) (Merck, Darmstadt, Germany) was visualized by UV light (254 nm) and spraying with cobalt chloride reagent (2% in sulfuric acid, 10%) and heating.

### 3.2. Biological Material

*Laurencia johnstonii* was collected by hand during June–July 2015 off the coast of Baja California Sur, Mexico (24°21′10.8′′ N, 110°16′58.8′′ W). A voucher specimen (code 13-003) was deposited at the Herbarium of the Laboratory of Marine Algae of the CICIMAR (Centro Interdisciplinario de Ciencias Marinas, Mexico). This seaweed was identified to genus and species level by Dr. Rafael Riosmena Rodríguez (Universidad Autónoma de Baja California Sur) using taxonomic keys for future reference.

### 3.3. Extraction and Isolation

Specimens of *Laurencia johnstonii* were cleaned of sand and epiphytes, and dried. Dried alga was crushed using a blender and extracted with EtOH for three days at 25 °C under gentle agitation. The EtOH was replaced (3 × 1200 mL), and the combined extracts were filtered through a Whatman no. 4 filter paper. Solvent was removed using a rotatory vacuum evaporator. 10.0 g of the resulting extract were chromatographed in Sephadex LH-20 (Sigma, St. Louis, MO, USA) (500 × 70 mm, CH_3_OH, 100%) to obtain five fractions. The active fraction SF3 was separated in flash Silicagel 0.2–0.5 mm (Sigma-Aldrich, St. Louis, MO, USA) (130 × 70 mm) using a stepwise gradient of *n*-hexane: ethyl acetate to obtain 7 fractions. Fractions SF3.1 (100% *n*-hexane) and SF3.2 (95% *n*-hexane) were separated on a normal phase open column (Silicagel, 0.2–0.5 mm, 300 × 50 mm, using a stepwise gradient of *n*-hexane: ethyl acetate) to yield pure compounds **5** (10.5 mg), **4** (7.2 mg) and **3** (45.9 mg) from SF3.1 and compounds **3** (10.2 mg), **2** (8.1 mg) and **1** (55.7 mg) from SF3.2. (Appendix A)

#### 3.3.1. Laurinterol (**1**)

White crystal; [α]${}_{D}^{25}$ +17 (*c* 0.15, CH_2_Cl_2_); HRESIMS *m*/*z* 293.0531 [M − H]^−^ (calc. C_15_H_18_O^79^Br, 293.0541), 295.0518 [M − H]^−^ (calc. C_15_H_18_O^81^Br, 295.0521) ^1^H NMR (500 MHz, CDCl_3_) δ 0.55 (1H, dd, *J* = 7.9, 4.8 Hz, H-12), 0.58 (1H, t, *J* = 4.6 Hz, H-12), 1.15 (1H, dt, *J* = 8.1, 4.3 Hz, H-3), 1.28 (1H, m, H-5), 1.32 (3H, s, H-13), 1.41 (3H, s, H-14), 1.66 (1H, dd, *J* = 12.3, 8.0 Hz, H-4), 1.95 (1H, tdd, *J* = 12.3, 8.1, 4.4, H-4), 2.09 (1H, dd, *J* = 13.2, 8.1 Hz, H-5), 2.29 (3H, s, H-15), 5.26 (1H, br, s, 7-OH), 6,61 (1H, s, H-8), 7.61 (1H, s, H-11); ^13^C NMR (125 MHz, CDCl_3_) δ 16.2 (C-12), 18.6 (C-13), 22.2 (C-14), 23.5 (C-15), 24.4 (C-3), 25.3 (C-4), 29.6 (C-2), 35.9 (C-5), 114.9 (C-10), 118.8 (C-8), 132.3 (C-11), 134.0 (C-6), 135.9 (C-9), 153.3 (C-7).

#### 3.3.2. Isolaurinterol (**2**)

Colorless amorphous solid; [α]${}_{D}^{25}$ −46 (*c* 0.14, CH_2_Cl_2_); HRESIMS *m*/*z* 293.0536 [M − H]^−^ (calc. C_15_H_18_O^79^Br, 293.0541), 295.0528 [M − H]^−^ (calc. C_15_H_18_O^81^Br, 295.0521) ^1^H NMR (500 MHz, CDCl_3_) δ 1.21 (3H, d, *J* = 7.0 Hz, H-12), 1.42 (1H, ddd, *J* = 12.8, 8.2, 6.6 Hz, H-5), 1.46 (3H, s, H-14), 1.59 (1H, dt, *J* = 12.9, 7.1, 7.1 Hz, H-4), 2.05 (1H, ddt *J* = 12.8, 8.5, 7.0, 7.0 Hz, H-3), 2.20 (1H, ddd, *J* = 13.0, 8.1 6.7 Hz, H-4), 2.31 (3H, s, H-15), 2.85 (1H, ddt, *J* = 9.1, 6.9, 6.9, 2.3, 2.3 Hz, H-5), 4.94 (1H, d, *J* = 2.2 Hz, H-13), 5.11 (1H, d, *J* = 2.2 Hz, H-13), 5.56 (1H, br, s, 7-OH), 6.73 (1H, s, H-8), 7.45 (1H, s, H-11); ^13^C NMR (125 MHz, CDCl_3_) δ 21.2 (C-14), 22.2 (C-15), 27.8 (C-12), 31.2 (C-4), 37.6 (C-3), 39.1 (C-5), 49.8 (C-1), 106.9 (C-13), 115.5 (C-10), 120.4 (C-8), 131.2 (C-11), 132.7 (C-6), 137.2 (C-9), 153.0 (C-7), 165.4 (C-2).

#### 3.3.3. Aplysin (**3**)

Colorless needles; [α]${}_{D}^{25}$ −49 (*c* 0.19, CH_2_Cl_2_); HRESIMS *m*/*z* 293.0551 [M − H]^−^ (calc. C_15_H_18_O^79^Br, 293.0541), 295.0529 [M − H]^−^ (calc. C_15_H_18_O^81^Br, 295.0524). ^1^H and ^13^C NMR data (Table 1).

#### 3.3.4. α-Bromocuparane (**4**)

White amorphous solid; [α]${}_{D}^{25}$ +21 (*c* 0.19, CH_2_Cl_2_); HREIMS [M − H]^+^
*m*/*z* 279.0744 (calc. for C_15_H_20_^79^Br, 279.0748), 282.0723 (calc. for C_15_H_20_^81^Br, 282.0728), [M − Br]^+^
*m*/*z* 201.1640 (calc. for C_15_H_21_, 201.1643); ^1^H NMR (500 MHz, CDCl_3_) δ 0.62 (3H, s, H-13), 1.09 (3H, s, H-12), 1.43 (3H, s, H-14), 1.96 (1H, m, H-5), 2.24 (1H, m, H-4), 2.27 (1H, m, H-5), 2.32 (3H, s, H-15), 2.52 (1H, ddt, *J* = 14.3, 9.3, 5.2, H-4), 4.07 (1H, t, *J* = 9.4 Hz, H-3), 7.09 (4H, m, H-7, H-8, H-10, H-11); ^13^C NMR (125 MHz, CDCl_3_) δ 20.7 (C-15), 20.9 (C-13), 22.4 (C-12), 25.0 (C-14), 33.1 (C-4), 36.4 (C-5), 47.4 (C-2), 48.4 (C-1), 62.0 (C-3), 127.3 (C-7, C-11), 128.1 (C-8, C-10), 135.4 (C-9), 143.9 (C-6).

#### 3.3.5. α-Isobromocuparane (**5**)

Colorless oil; [α]${}_{D}^{25}$ +37 (*c* 0.14, CH_2_Cl_2_); HREIMS [M]^+^
*m*/*z* 280.0826 (calc. for C_15_H_21_^79^Br, 280.0827), 282.0801 (calc. for C_15_H_21_^81^Br, 282.0806), [M − Br]^+^
*m*/*z* 201.1649 (calc. for C_15_H_21_, 201.1643); ^1^H NMR (500 MHz, CDCl_3_) δ 0.65 (3H, s, H-13), 1.10 (3H, s, H-12), 1.28 (3H, s, H-14), 1.61 (1H, m, H-5), 2.20 (1H, m, H-4), 2.33 (3H, s, H-15), 2.51 (1H, dtd, *J* = 14.2, 9.5, 6.5 Hz, H-4), 2.71 (1H, td, *J* = 12.6, 6.4 Hz, H-5), 4.46 (1H, t, *J* = 9.2 Hz, H-3), 7.12 (1H, d, *J* = 8.1 Hz, H-10), 7.13 (1H, d, *J* = 8.1 Hz, H8), 7.27 (2H, d, *J* = 8.1, H-7, H-11); ^13^C NMR (125 MHz, CDCl_3_) δ 20.6 (C-15), 20.8 (C-13), 22.2 (C-12), 25.3 (C-14), 31.2 (C-4), 33.5 (C-5), 47.7 (C-2), 48.4 (C-1), 63.6 (C-3), 126.5 (C-7, C-11), 128.5 (C-8, C-10), 135.4 (C-9), 143.0 (C-6).

### 3.4. Transformaction of Natural Sesquiterpenes ***1*** and ***2***

Laurinterol (**1**, 8.0 mg, 0.027 mmol) was dissolved in ethyl ether (5 mL) and bromine (100 μL, 1.95 mmol) was added. The reaction was left under magnetic stirring for 15 min at room temperature, after which the solvent was evaporated in vacuo. The reaction mixture was purified on a silica gel 0.2–0.5 mm (Sigma-Aldrich, St. Louis, MO, USA) column (10 Ø × 50 mm) using a step gradient of *n*-Hex/EtOAc (100:0–95:5) to yield 8-bromoaplysin (**6**, 1.49 mg, 14.8%) and **7** (3.45 mg, 29.6%). Similarly, isolaurinterol (**2**, 5.0 mg, 0.017 mmol) was treated with bromine (100 μL, 1.95 mmol) in the same experimental conditions. After evaporation of the solvent, the reaction mixture was purified on a silica gel column (10 Ø × 50 mm; step gradient of *n*-Hex/EtOAc (100:0–95:5)) to give compound **6** (1.32 mg, 23.5%), **8** (0,38 mg, 6%) and **9** (1.81 mg, 27.3%).

Laurinterol (**1**, 10.0 mg, 0.034 mmol) was dissolved in ethyl ether (5 mL) and HBr (10 μL, 48%) was added. The reaction was left under magnetic stirring for 15 min at room temperature, after which the solvent was evaporated in vacuo. Quantitative conversion was observed to aplysin (**3**) which was purified by crystallization in *n*-hexane.

#### 3.4.1. 8-Bromoaplysin (**6**)

Colorless oil; [α]${}_{D}^{20}$ −30 (*c* 0.12, CHCl_3_); IR υ_max_ 3264, 1652, 1315, 1103, 1022, 904 cm^−1^; HREIMS [M]^+^
*m*/*z* 371.9711 (calc. for C_15_H_18_O^79^Br_2_, 371.9724), 373.9702 (calc. for C_15_H_18_O^79^Br^81^Br, 373.9704), 375.9677 (calc. for C_15_H_18_O^81^Br_2_, 375.9683); ^1^H and ^13^ C NMR data (Table 1).

#### 3.4.2. 3α-Bromojohnstane (**7**)

Colorless oil; [α]${}_{D}^{20}$ −35 (*c* 0.19, CHCl_3_); IR υ_max_ 3259, 2362, 2158, 2031, 1975, 906, 800 cm^−1^; HREIMS [M]^+^
*m*/*z* 449.8840 (calc. for C_15_H_17_O^79^Br_3_, 449.8829), 451.8800 (calc. for C_15_H_17_O^79^Br_2_^81^Br, 451.8809), 453.8798 (calc. for C_15_H_17_O^79^Br^81^Br_2_, 453.8789), 455.8758 (calc. for C_15_H_17_O^81^Br_3_, 455.8768); ^1^H and ^13^ C NMR data (Table 2).

#### 3.4.3. 8,10-Dibromoisoaplysin (**8**)

Colorless oil; [α]${}_{D}^{20}$ −17 (*c* 0.04, CHCl_3_); HREIMS [M]^+^
*m*/*z* 449.8836 (calc. for C_15_H_17_O^79^Br_3_, 449.8829), 451.8815 (calc. for C_15_H_17_O^79^Br_2_^81^Br, 451.8809), 453.8783 (calc. for C_15_H_17_O^79^Br^81^Br_2_, 453.8789), 455.8764 (calc. for C_15_H_17_O^81^Br_3_, 455.8768).

#### 3.4.4. 8,10-Dibromoaplysinol (**9**)

Colorless oil; [α]${}_{D}^{20}$ −6 (*c* 0.08, CHCl_3_); HREIMS [M]^+^
*m*/*z* 387.9680 (calc. for C_15_H_18_O_2_^79^Br_2_, 387.9674), 389.9667 (calc. for C_15_H_18_O_2_^79^Br^81^Br, 389.9653), 391.9639 (calc. for C_15_H_18_O_2_^81^Br_2_, 391.9633).

### 3.5. Cell Culture

The amoeba strain used in this study was the type strain: *Acanthamoeba castellanii* Neff (ATCC 30010) which was axenically grown in PYG medium (0.75% (*w*/*v*) proteose peptone, 0.75% (*w*/*v*) yeast extract and 1.5% (*w*/*v*) glucose) containing 40 μg/mL of gentamicin (Biochrom AG, Cultek, Granollers, Barcelona, Spain).

The murine macrophages J774A.1 (ATCC# TIB-67) cell line was cultured in RPMI 1640 medium supplemented with 10% fetal bovine serum at 37 °C and 5% CO_2_ atmosphere, was used for the cytotoxicity assays.

### 3.6. Anti-Acanthamoeba Activity

To evaluate the biological activity of the fractions and molecules, the anti-*Acanthamoeba* assays were determined using the alamarBlue^®^ method as previously described [15,16]. Firstly, *Acanthamoeba* strain was seeded in triplicate on a 96-well microtiter plates with 50 μL from a stock solution of 10^4^ cells/mL. Amoebae were left to adhere to the bottom of the well for 15 min, which was checked using a Leika DMIL inverted microscope (Leika, Wetzlar, Germany). After that, 50 μL of serial dilutions of the molecules were added to the wells (1% DMSO was used to dissolve the highest dose of the compounds with no effects on the parasites). As a control, we use chlorhexidine (chlorhexidine digluconate; Alfa Aesar), which is a standard antiseptic belonging to the biguanide family of antiseptics that is commonly used in contact lens maintenance solutions; and voriconazole (Sigma) is an inhibitor of ergosterol synthesis that has been proven previously to be highly effective against clinical strains of *Acanthamoeba* [26,27]. Finally, the alamarBlue Reagent^®^ (Life Technologies, Madrid, Spain) was placed into each well at an amount equal to 10% of the final volume. Then, plates containing alamarBlue^®^ were incubated for 96 h at 26 °C with a slight agitation.

### 3.7. Cytotoxicity Test

The cytotoxicity assay was carried out using the alamarBlue^®^ method as previously described [15,16]. Firstly, the macrophages were seeded in triplicate on a 96-well microtiter plates with 50 μL from a stock solution of 2 × 10^5^ cells/mL. Macrophages were left to adhere to the bottom of the well for 15 min, which was checked using a Leika DMIL inverted microscope (Leika, Wetzlar, Germany). After that, 50 μL of serial dilutions of the molecules were added to the wells. Finally, the alamarBlue Reagent^®^ (Life Technologies, Madrid, Spain) was placed into each well at an amount equal to 10% of the final volume. Then, plates containing alamarBlue^®^ were incubated for 24 h at 37 °C in presence of CO_2_ at 5%.

### 3.8. Statical Analysis

Briefly, the test plates were analyzed, during an interval of time between 24 and 96 h, on an EnSpire^®^ Multimode Plate Reader (Perkin Elmer, Madrid, Spain) using fluorescence, a test wavelength of 570 nm and a reference wavelength of 630 nm. The percentage of the growth inhibition, 50% inhibitory concentration (IC_50_ or CC_50_) was calculated by linear regression analysis with 95% confidence limits using Sigma Plot 12.0 statistical analysis software (Systat Software). All experiments were performed three times, and the mean values were also calculated. A paired two-tailed t-test was used for analysis of the data. Values of *p* < 0.05 were considered significant.

## 4. Conclusions

To the best of our knowledge, red algae have never been evaluated as a potential source of amoebicidal agents. Furthermore, this is the first time that the anti-*Acantamoeba* activity of the brominated sesquiterpenes α-bromocuparane (**4**) and α-isobromocuparane (**5**), isolated from *Laurencia johnstonii*, is reported. On the other hand, after chemical modification of the inactive metabolites laurinterol (**1**) and isolaurinterol (**2**), a set of brominated derivatives, **6**–**9**, that substantially improve the activity of the natural sesquiterpenes have been obtained. The addition of a second bromine atom in the aromatic ring of **6**–**9** seems to be relevant to increase the antiamoeboid activity. Whereas all obtained derivatives possess a laurane-type framework, the most active compound, 3α-bromojohnstane (**7**), possesses a different rearranged carbon skeleton, johnstane, never found among natural sesquiterpene metabolites isolated from *Laurencia* species so far. Our results suggest that *Laurencia*-based brominated sesquiterpenes could be a potential source of novel therapeutic agents against *Acanthamoeba* in the near future.

## Figures and Tables

**Figure 1 marinedrugs-16-00443-f001:**
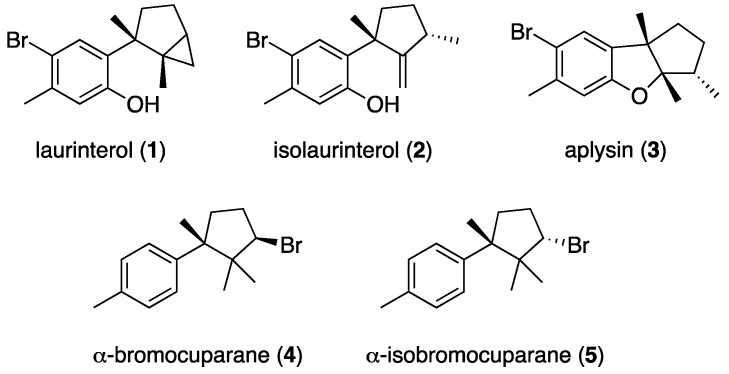
Structures of natural sesquiterpenes (**1**)–(**5**) isolated from *Laurencia jonhstonii*.

**Figure 2 marinedrugs-16-00443-f002:**
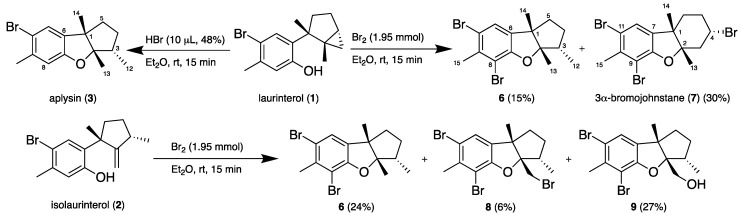
Bromination reaction of natural laurinterol (**1**) and isolaurinterol (**2**).

**Figure 3 marinedrugs-16-00443-f003:**
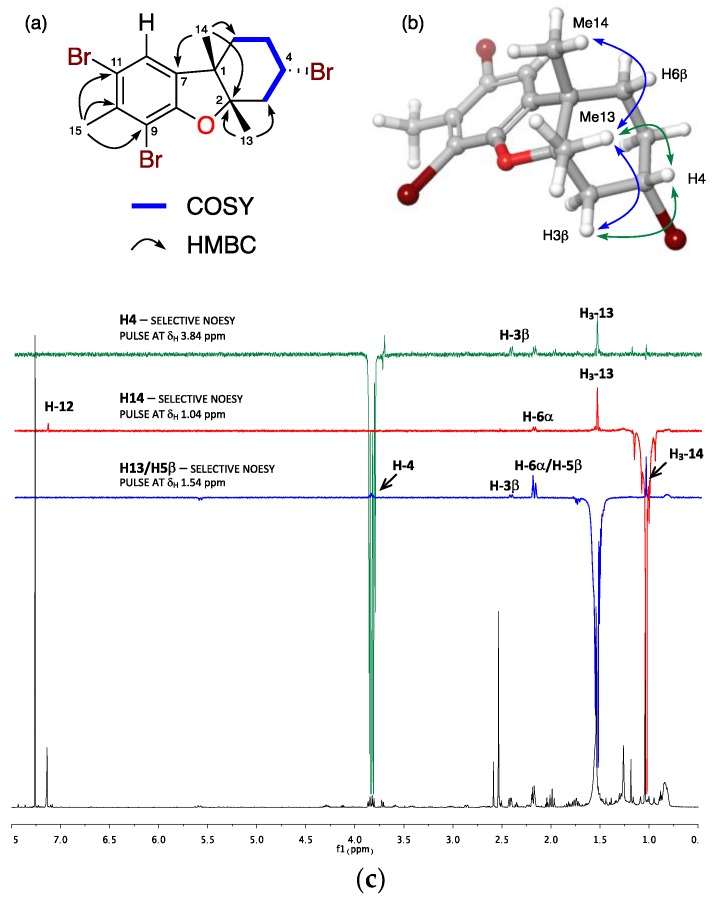
(**a**) Selected COSY and HMBC correlations, (**b**) key-NOE correlations and (**c**) 1D-NOE experiments of 3α-bromojohnstane (**7**).

**Figure 4 marinedrugs-16-00443-f004:**
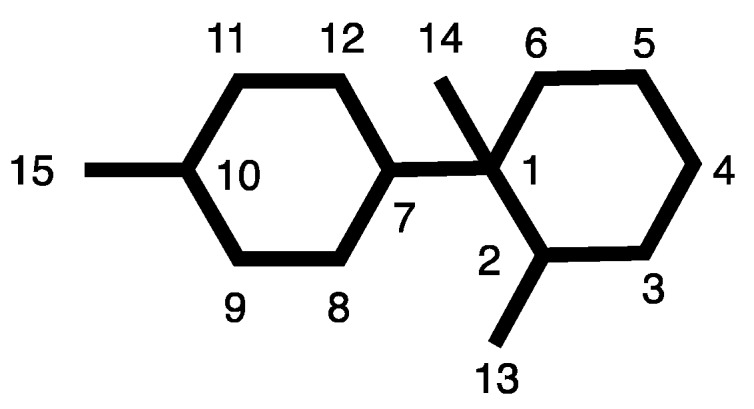
Johnstane carbon skeleton and numbering system.

**Table 1 marinedrugs-16-00443-t001:** ^1^H and ^13^C NMR data for aplysin (**3**) and (**6**) (600 MHz, 150 MHz, CDCl_3_).

Position	Aplysin (3)		6	
	δ_C_, Type	δ_H_ (*J* in Hz)	δ_C_, Type	δ_H_ (*J* in Hz)
1	54.3, C		56.0, C	
2	99.8, C		100.5, C	
3	46.0, CH	1.75, ddd (12.8, 6.9, 6.5)	46.4, CH	1.79, ddd (13.3, 6.5, 6.5)
4	31.1, CH_2_	1.62, m	31.3, CH_2_	1.65, m
		1.15, m		1.15, m
5	42.5, CH_2_	1.85, dd (6.4, 12.1)	42.7, CH_2_	1.86, dd (12.1, 6.2)
		1.58, d (6.2)		1.64, ddd (12.1, 6.2, 6.2)
6	136.2, C		136.2, C	
7	158.1, C		156.0 ^1^, C	
8	110.8, CH	6.59, s	105.1, C	
9	136.9, C		136.0, C	
10	113.9, C		114.1, C	
11	126.5, CH	7.14, s	125.4, CH	7.11, s
12	13.0, CH_3_	1.10, d (6.8)	13.3, CH_3_	1.14, d (6.8)
13	23.1, CH_3_	1.30, s	20.4, CH_3_	1.34, s
14	23.3, CH_3_	1.29, s	23.2, CH_3_	1.32, s
15	29.9, CH_3_	2.30, s	23.5, CH_3_	2.49, s

^1^ Chemical shift deduced from the HMBC experiment.

**Table 2 marinedrugs-16-00443-t002:** NMR data for 3α-bromojohnstane (**7**) (600 MHz, 150 MHz, CDCl_3_).

Position	3α-Bromojohnstane (7)	
	δ_C_, Type	δ_H_ (*J* in Hz)
1	48.1, C	
2	92.3, C	
3	47.6, CH_2_	β: 2.42 dd (3.6, 12.9)
		α: 1.99 dd (12.6, 12.9)
4	45.8, CH	3.84 dddd (3.6, 3.6, 12.3, 12.6)
5	34.2, CH_2_	β: 2.18 dddd (3.2, 3.5, 3.6, 12.9)
		α: 1.76 dddd (3.5, 12.3, 12.3, 12.6)
6	32.8, CH_2_	β: 1.54 ddd (3.5, 12.3, 12.9)
		α: 2.18 ddd (3.2, 3.5, 12.9)
7	136.3, C	
8	155.3, C	
9	107.6, C	
10	136.6, C	
11	115.5, C	
12	124.6, CH	7.14 s
13	19.7, CH_3_	1.54 s
14	26.7, CH_3_	1.04 s
15	23.3, CH_3_	2.53 s

**Table 3 marinedrugs-16-00443-t003:** Effect of *Laurencia johnstonii* ethanolic extract and **1**–**9** against *Acanthamoeba castellanii* Neff (IC_50_) and murine macrophage J774.A1 (CC_50_). * Reference compounds.

Sample	IC_50_ (µg/mL)	IC_50_ (µM)	CC_50_ (µg/mL)
Crude extract	125.14 ± 4.5		n/d
**1**	>100		23.65 ± 2.3
**2**	>100		7.25 ± 0.7
**3**	>100		323.69 ± 12.0
**4**	90.674 ± 1.529	322.41 ± 5.44	33.04 ± 4.2
**5**	64.251 ± 3.492	228.46 ± 12.42	85.34 ± 10.9
**6**	24.559 ± 1.105	65.64 ± 2.95	32.880 ± 3.125
**7**	18.804 ± 0.198	41.51 ± 0.44	62.341 ± 2.589
**8**	22.818 ± 1.896	50.37 ± 4.19	70.365 ± 3.245
**9**	29.937 ± 2.918	76.74 ± 7.48	74.931 ± 2.769
Chlorhexidine *	1.526 ± 0.45	3.02 ± 0.89	6.64 ± 0.35
Voriconazole *	0.33 ± 0.1	0.94 ± 0.29	2.64 ± 0.27

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
