# Peer review of "Anti-Acanthamoeba Activity of Brominated Sesquiterpenes from Laurencia johnstonii"

_marinedrugs, 2018, doi:10.3390/md16110443_

Reviewer 1 Report

The manuscript "Anti-Acanthamoeba activity of brominated sesquiterpenes from Laurencia johnstonii" described thethe anti-Acantamoeba activity of an extract of Laurencia johnstonii. 

The paper is well written and well organized. The results are clearly presented.  For these reasons the manuscript can be published in the journal.

Authors described new Laurencia-based brominated sesquiterpenes as potential lead compounds  for the development of novel therapeutic agents against Acanthamoeba, highlighting as the introduction  of bromine atoms in the aromatic ring led to an improvment in the biological activity.

Even if the biological activty showed by the brominated sesquiterpenes against Acanthamoeba castellanii Neff  is not excellent (the most active compound displayed an IC50 value of 41,51 ÎĽM), the results allow to deepen the SAR of these compounds.

Details:

Page 9: author must add the chemical names after compound 6...compound 7...

Fig 4: the image is confusing, it should be better to use only a type of chemical structure for molecule.

Author Response

The manuscript "Anti-Acanthamoeba activity of brominated sesquiterpenes from Laurencia johnstonii" described thethe anti-Acantamoeba activity of an extract of Laurencia johnstonii. 

The paper is well written and well organized. The results are clearly presented.  For these reasons the manuscript can be published in the journal.

Authors described new Laurencia-based brominated sesquiterpenes as potential lead compounds  for the development of novel therapeutic agents against Acanthamoeba, highlighting as the introduction  of bromine atoms in the aromatic ring led to an improvment in the biological activity. 

Even if the biological activty showed by the brominated sesquiterpenes against Acanthamoeba castellanii Neff  is not excellent (the most active compound displayed an IC50 value of 41,51 ÎĽM), the results allow to deepen the SAR of these compounds.

Response:

Thank you for the valuable comments to improve the manuscript. As suggested, we have included additional discussion regarding the SAR of the compounds.

Details:

Page 9: author must add the chemical names after compound 6...compound 7...

Response:

The chemical names of compounds have been included. The names have also been included in the Supporting Information document.

Fig 4: the image is confusing, it should be better to use only a type of chemical structure for molecule.

Response:

Due to the lack of data to confirm the proposed mechanism of compound 7, figure 4 have been omitted.

Reviewer 2 Report

In this paper the Authors isolate some sesquiterpene bromurated compounds from Laurencia johnstonii. After isolation and full purification and full structural assignment, the authors proceed with further bromuration of the obtained compounds in order to obtain new analogues which are good against Acantoamoeba. The structural assignment is conducted in a proper way, and the combination of the chemical physical approaches (NMR and MS) is state-of-art. 

Authors should motivate in a clear cut way the further reaction on the obtained compounds, and maybe argue why bromuration among all possible reactions.

I totally disagree however on the mechanism of formation of the new sesquiterpene with johnstane skeleton. There is no evidence for such a mechanisms which is also poorly explained and also at least to my comprehension contains great mistakes in Chemistry. I see positive charges leaving the formulas and formula becoming carbocation. please delete this part.

Author Response

Response to Reviewer 2 Comments

In this paper the Authors isolate some sesquiterpene bromurated compounds from Laurencia johnstonii. After isolation and full purification and full structural assignment, the authors proceed with further bromuration of the obtained compounds in order to obtain new analogues which are good against Acantoamoeba. The structural assignment is conducted in a proper way, and the combination of the chemical physical approaches (NMR and MS) is state-of-art. 

Authors should motivate in a clear cut way the further reaction on the obtained compounds, and maybe argue why bromuration among all possible reactions.

Response:

Thank you for the valuable comments to improve the manuscript. As suggested, we have included details for selection of bromuration reactions.

I totally disagree however on the mechanism of formation of the new sesquiterpene with johnstane skeleton. There is no evidence for such a mechanisms which is also poorly explained and also at least to my comprehension contains great mistakes in Chemistry. I see positive charges leaving the formulas and formula becoming carbocation. please delete this part

Response:

We agree about the lack of conclusive data for to confirm the proposed mechanism to form compound 7, so, meanwhile we are able to provide additional information, figure 4 has been omitted.

Round  2

Reviewer 2 Report

paper is now fine with me. 

there has been several editing and revising; therefore I trust that the paper is now acceptable for publication as it is